# Dyn-VPP: Video Prediction Policy Optimization for Improved Visual Dynamics

**Zirui Ge** [* 1]   **Pengxiang Ding** [* 1 2]   **Baohua Yin** [* 3]   **Yemin Wang** [4]   **Qishen Wang** [5]   **Zhiyong Xie** [6]   **Hengtao Li** [7]
**Runze Suo** [8]   **Wenxuan Song** [9]   **Han Zhao** [1 2]   **Shangke Lyu** [10]   **Haoang Li** [9]   **Ran Cheng** [11]   **Cheng Chi** [12]
**Huibin Ge** [1]   **Yaozhi Luo** [1]   **Donglin Wang** [2]

## Abstract

Video action models are a promising foundation for Vision–Language–Action (VLA) because they can learn rich **visual dynamics** directly from video. However, likelihood-oriented training of diffusion predictors emphasizes globally plausible futures and does not guarantee **precision-critical visual dynamics** needed for manipulation, so small prediction errors can be amplified by downstream policies. We propose **Dyn-VPP**, a post-training framework that casts multi-step denoising as policy optimization and aligns predicted future latents with **expert visual dynamics** via a verifiable terminal reward, without modifying any architecture. This enables explicit optimization of dynamics signals that are not captured by likelihood-only training. As a result, **Dyn-VPP** yields more accurate **visual dynamics** and improves downstream task execution. Experiments across diverse simulated and real-world manipulation settings show improved dynamics consistency and consistently higher task success.

## 1. Introduction

Video action models are a promising foundation for Vision–Language–Action (VLA) systems because they can learn rich **visual dynamics** directly from video (Hu et al., 2025b).

This offers two practical advantages: (i) visual dynamics can be learned from large-scale video corpora without action labels, enabling scalable pretraining; and (ii) the learned dynamics can serve as a strong prior that improves data efficiency when learning action-conditioned behaviors from limited robot demonstrations, since manipulation-relevant state evolution (e.g., pose changes and contact progression) is reflected in the video.

A central question is how to represent and expose these dynamics for downstream control. Early approaches often predict **pixel-level** goal images, which are intuitive but typically expensive to generate via multi-step denoising and may include redundant appearance details that are weakly coupled to action selection (Du et al., 2023; Black et al., 2024; Bu et al., 2024). Consequently, recent methods increasingly favor **latent-level** representations that encode dynamics more compactly and can be extracted from early denoising steps in diffusion models (e.g., via early-exit features), enabling efficient inference while emphasizing geometry- and relation-centric cues (Wu et al., 2024; Wen et al., 2024; Tian et al., 2025; Hu et al., 2025a; Liu et al., 2025b).

Despite these advances, existing pipelines still struggle to provide precision-critical visual dynamics for manipulation. Diffusion-based predictors are usually trained with likelihood-surrogate objectives (e.g., ELBO-style formulations) (Kingma et al., 2021) that emphasize global plausibility under the data distribution, but do not directly optimize the accuracy of precision-critical factors (Black et al., 2023). As a result, predicted representations can be globally coherent yet contain subtle errors; when consumed by a downstream policy, these errors can be amplified near decision boundaries, leading to inaccurate actions and compounding failures over time.

To address this objective mismatch, we introduce **Dyn-VPP**, a video prediction policy optimization method for improved visual dynamics (Figure 1). Our key idea is to interpret multi-step denoising as a sequential decision process (Black et al., 2023): starting from a noisy latent, each denoising step takes an action (the denoiser update) that induces a transition defined by the sampler dynamics, and the process terminates at a predicted future latent representing the

---

[*]Equal contribution [§]Project leader [1]Zhejiang University, Hangzhou, China [2]Westlake University, Hangzhou, China [3]University of Sussex, United Kingdom [4]Xiamen University, Xiamen, China [5]Tianjin University, Tianjin, China [6]Wuhan University, Wuhan, China [7]Hebei University of Technology, Tianjin, China [8]Fudan University, Shanghai, China [9]HKUST (Guangzhou), Guangzhou, China [10]Nanjing University, Nanjing, China [11]McGill University, Montréal, Canada [12]Beijing Academy of Artificial Intelligence, Beijing, China. Correspondence to: Yaozhi Luo <luoyz@zju.edu.cn>, Donglin Wang <wangdonglin@westlake.edu.cn>.

*Proceedings of the 43rd International Conference on Machine Learning*, Seoul, South Korea. PMLR 306, 2026. Copyright 2026 by the author(s).

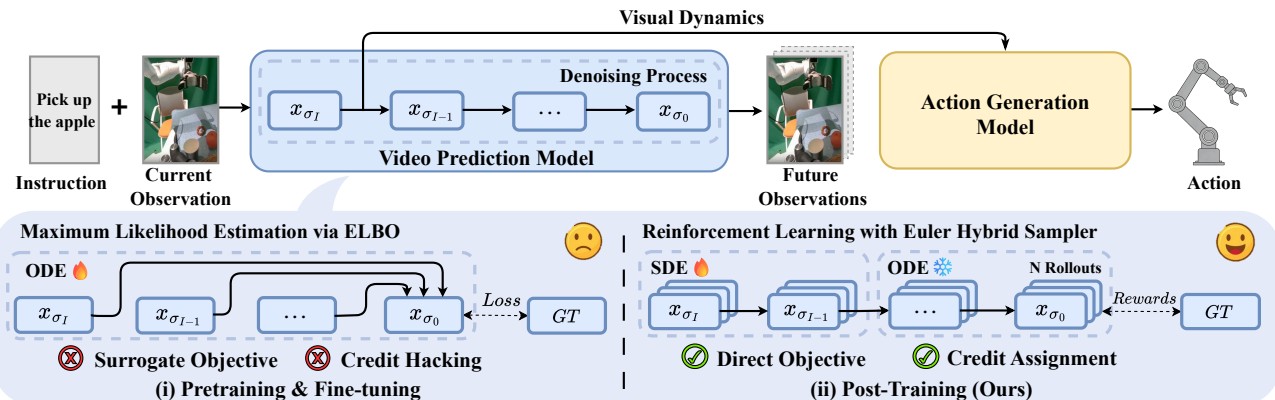

*Figure 1.* Overview of **Dyn-VPP**. Our post-training framework introduces reinforcement learning from verified rewards in place of the surrogate objective in Video Action Models, enabling direct optimization of task-specific goals in training video prediction model (VPM). This approach improves the accuracy of VPM's predictive visual representations, leading to enhanced action generation and task performance. Notably, our method demonstrates significant improvements not only in simulated environments but also in real-world scenarios, showcasing its robustness and versatility across diverse settings.

model's forecast. We then optimize this denoising policy to maximize a terminal reward that measures consistency with expert visual dynamics, computed by encoding the ground-truth future sequence in the same latent space. In this way, **Dyn-VPP** enables explicit optimization of the dynamics signals that are not captured by likelihood-only training.

**Dyn-VPP** instantiates this idea with three design choices tailored for learning **improved visual dynamics**: (1) an **Euler Hybrid sampler** that introduces SDE-style stochasticity only at the first denoising step (Karras et al., 2022), yielding a tractable Gaussian transition for low-variance policy-gradient estimation via reparameterization, while mitigating credit assignment issues and preserving the temporal coherence of the remaining deterministic denoising process; and (2) **GRPO** (Xue et al., 2025) with a verifiable, non-adversarial reward that combines $L_1$ distance and cosine similarity to align the predicted and expert visual dynamics. With these designs, Dyn-VPP yields more accurate **visual dynamics** and thereby improves downstream task execution. Extensive experiments across diverse simulated and real-world scenarios show that **Dyn-VPP** consistently enhances manipulation performance and generalization.

Our contributions are summarized as:

- We highlight an objective mismatch in diffusion-based video prediction: likelihood-oriented training does not guarantee **improved visual dynamics** required by manipulation policies.

- We propose **Dyn-VPP**, a post-training framework that casts diffusion denoising as policy optimization to directly improve **visual dynamics** with a terminal reward derived from **expert visual dynamics**.

- We design an **Euler Hybrid sampler** that incorporates

SDE-style Euler-Ancestral sampling that enable stable and efficient optimization by balancing exploration, temporal coherence, and credit assignment.

## 2. Revisiting the Video Action Model

### 2.1. Existing Paradigms.

A typical video action model comprises two modules: a video prediction model (VPM) and an action generation model (AGM). The VPM anticipates a representation of the future conditioned on the current observation and a language instruction, while the AGM plays the role of an inverse-dynamics component that converts the predicted future representation—optionally together with additional conditioning signals—into an executable action sequence. A key design choice concerns the representation produced by the VPM, which leads to two dominant families of video action models:

**Pixel-level approaches.** In pixel-level formulations (Du et al., 2023; Black et al., 2024; Bu et al., 2024), the predicted future takes the form of an explicit goal observation, such as a future image or a short video clip. This design is attractive for its interpretability, since the model's intended outcome can be directly visualized. However, predicting in pixel space is often unnecessary for control: it forces the model to synthesize high-frequency appearance details (e.g., texture and illumination) that are only weakly coupled to action selection, introducing redundancy and potentially allowing prediction errors in visual details to propagate into downstream decision making.

**Latent-level approaches.** In latent-level formulations (Wu et al., 2024; Wen et al., 2024; Tian et al., 2025; Hu et al., 2025a; Liu et al., 2025b), the VPM outputs a compact inter-

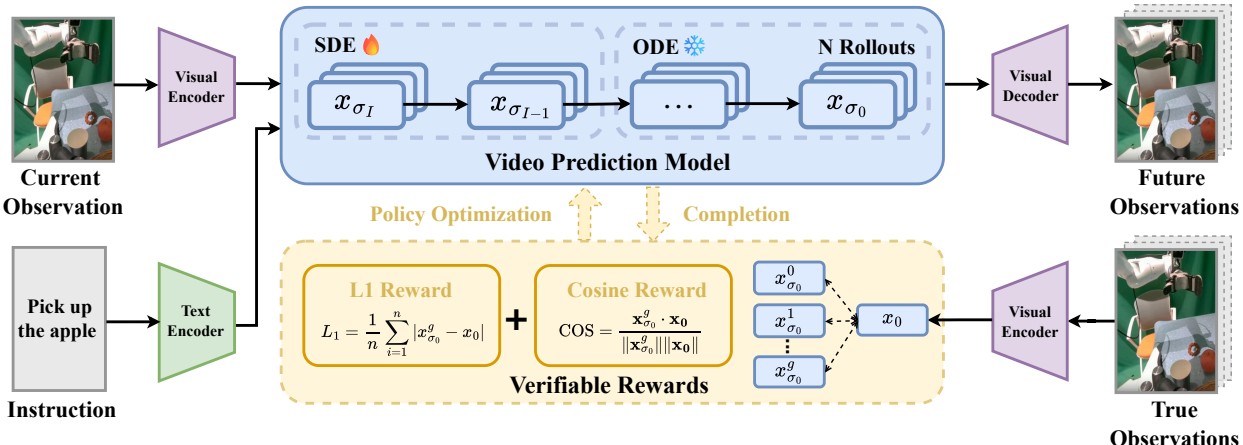

*Figure 2.* Overview of the **Dyn-VPP** training paradigm. In the pre-training stage, the video prediction model (VPM) and action generation model (AGM) are trained on expert demonstrations. In the policy optimization stage, the VPM generates future latents via a hybrid denoising process (SDE at the first step, ODE thereafter); verified rewards are computed by comparing predicted and expert latents, and the VPM is optimized with GRPO to improve precision-critical visual dynamics for downstream action generation.

mediate representation rather than decoded pixels, and the action generator conditions primarily on this representation. For diffusion-based VPMs in particular, the relevant future representation is commonly extracted from intermediate denoising features rather than from fully generated images. Such latent representations tend to suppress nuisance visual variability while retaining task-relevant dynamics and affordances, thereby enabling more efficient and robust control (Wu et al., 2024; Wen et al., 2024; Tian et al., 2025; Hu et al., 2025a). Moreover, intermediate denoising features can often be obtained using only the earliest denoising step(s), substantially reducing inference cost and mitigating the runtime bottleneck that would otherwise hinder real-time robotic deployment. Consequently, latent-level designs have become the prevailing choice in practice. And in this work, we also adopt this paradigm and conduct our analyses under this setting.

## 2.2. Instantiation of Latent-level Paradigm.

Here, we follow the standard latent-level video action modeling paradigm (Hu et al., 2024) to instantiate our approach in a concrete form. Specifically, we consider a dataset $\mathcal{D} = \{(o, l, a, N)_i\}_{i=1}^M$ consisting of expert demonstrations, where each transition contains a current observation $o$, a language instruction $l$, a ground-truth action sequence $\mathbf{a}$, and a target future video clip $N$. Given a sample from $\mathcal{D}$, we form the video-model condition $c_v = (o, l)$. The VPM produces a predicted future representation $\hat{x}_0$ and an intermediate visual feature $\hat{h}$. Subsequently, the AGM generates an action sequence $\hat{a}_0$ conditioned on $c_a = (\hat{h}, l)$:

$$(\hat{x}_0, \hat{h}) = \text{VPM}_\theta(c_v), \qquad \hat{a}_0 = \text{AGM}_\theta(c_a). \quad (1)$$

**Details of VPM.** We instantiate VPM with Stable Video

Diffusion (SVD) (Blattmann et al., 2023), a latent video diffusion model built on denoising diffusion models (Ho et al., 2020) with EDM-style design and sampling (Karras et al., 2022). Let $N$ denote an expert demonstration video clip (a sequence of Observations) aligned with condition $c_v$. We encode $N$ into the latent space using a pretrained VAE encoder $\mathcal{E}$, yielding the clean latent $x_0 = \mathcal{E}(N)$, which serves as the target future representation. Following EDM, the forward noising process directly adds Gaussian noise at a continuous noise level $\sigma$:

$$x_\sigma = x_0 + \sigma\epsilon, \quad \epsilon \sim \mathcal{N}(0, \mathbf{I}). \quad (2)$$

The conditional denoiser $D_{\theta_v}(x_{\sigma_i}, \sigma_i, c_v)$ predicts the clean latent $x_0$ from a noisy sample $x_{\sigma_i}$ under guidance of $c_v$. The supervised training objective is:

$$\mathcal{L}_{Video} = \mathbb{E}_{x_0, \epsilon, \sigma_i} \left\| D_{\theta_v}(x_{\sigma_i}, \sigma_i, c_v) - x_0 \right\|^2 \quad (3)$$

At inference time, given a decreasing noise schedule $\sigma_I > \cdots > \sigma_0 = 0$, we generate the predicted future representation by numerically integrating the probability-flow ODE with Euler Discrete solver:

$$x_{\sigma_{i-1}} = x_{\sigma_i} + (\sigma_{i-1} - \sigma_i) \frac{x_{\sigma_i} - D_{\theta_v}(x_{\sigma_i}, \sigma_i, c_v)}{\sigma_i}, \quad (4)$$

Following (Hu et al., 2024), once the model is trained, we extract the predictive visual representation $\hat{h}$ from the multi-layer hidden states of $D_{\theta_v}$ at the first denoising step. This representation captures high-level spatiotemporal structure and task-relevant dynamics, and is used to condition the AGM.

**Details of AGM.** The AGM generates an action sequence $\mathbf{a}_0$ conditioned on $c_a = (\hat{h}, l)$. We adopt a diffusion policy

(Chi et al., 2025) as the action head and use a DDIM-style denoising process (Song et al., 2020a). Given a ground-truth action sequence $\mathbf{a}_0$, the forward noising process constructs a noisy action at diffusion step $k$:

$$a_k = \sqrt{\bar{\beta}_k}\, a_0 + \sqrt{1 - \bar{\beta}_k}\, \epsilon, \quad \epsilon \sim \mathcal{N}(0, \mathbf{I}), \quad (5)$$

where $\bar{\beta}_k$ is the (cumulative) noise coefficient at step $k$. The action denoiser $D_{\theta_a}$ is trained to reconstruct $\mathbf{a}_0$ by minimizing:

$$\mathcal{L}_{Action} = \mathbb{E}_{a_0,\,k} \left\| D_{\theta_a}(a_k, k, c_a) - a_0 \right\|^2. \quad (6)$$

### 2.3. Limitation Analysis.

However, diffusion-based VPMs are optimized with distribution-level objectives (e.g., likelihood surrogates), rather than with a direct objective for high-precision, control-relevant future representations. This creates an objective mismatch: improving likelihood does not necessarily yield representations that are reliable for downstream action generation. In practice, video diffusion models are trained by approximately maximizing the data likelihood, commonly through a variational bound such as the ELBO (or an equivalent surrogate loss) (Kingma et al., 2021; Ho et al., 2020). This introduces approximation at multiple levels: the optimized objective is only a proxy for the true likelihood, and the finite-step stochastic sampling procedure can further introduce error. As a result, the produced future representation is not guaranteed to be precise in the specific state variables that matter for control. Consequently, VPM outputs can be globally coherent yet subtly incorrect in either pixel space or latent space, exhibiting small errors in object pose, contact timing, or fine-grained spatial relations (e.g., slight misalignment, small gaps, or off-by-one-frame contact events). When this representation conditions an AGM, these small discrepancies can be amplified. Near decision boundaries, minor representational errors can flip action choices (e.g., whether to close a gripper, which side to route around an obstacle, or when to initiate contact). Once an incorrect action is executed, the environment state can diverge further, causing compounding errors over time and ultimately yielding degraded task success.

### 2.4. Our Solution

As illustrated in figure 2, a natural way to resolve the objective mismatch above is to directly optimize the VPM for the accuracy of its generated visual dynamics, rather than relying solely on likelihood-based surrogates. Concretely, we treat the multi-step denoising process as a stochastic generator, define a reward by comparing the final prediction against a task-specific target (e.g., ground-truth representations or oracle-derived features), and update the model using policy-gradient–style optimization (Black et al., 2023). This

objective penalizes subtle yet control-critical errors (e.g., inaccuracies in pose, contact, and spatial relations), thereby narrowing the gap between generative training and action-centric deployment. In the following sections, we detail how to implement this framework.

## 3. Method

We present **Dyn-VPP**, a post-training framework that optimizes video prediction policies for improved visual dynamics. Specifically, we (1) formulate video denoising during generation as a Markov Decision Process (MDP); (2) introduce a stochastic sampler that makes the denoising trajectory amenable to likelihood-based optimization; and (3) apply a reinforcement learning algorithm with a hybrid denoising sampler, enabling the model to directly align visual dynamics. By improving the fidelity of learned visual dynamics, **Dyn-VPP** enables more accurate and reliable video predictions, which in turn transfer to downstream policies to support more effective action generation

### 3.1. Problem formulation

Following DDPO (Black et al., 2023), we model the denoising process of a diffusion-based VPM as a Markov Decision Process (MDP) $\mathcal{M} = (\mathcal{S}, \mathcal{A}, \rho_0, \boldsymbol{P}, \boldsymbol{R})$, which consists of five key components: the state space $\mathcal{S}$, action space $\mathcal{A}$, initial state distribution $\rho_0$, transition kernel $\boldsymbol{P}$, and reward function $\boldsymbol{R}$. Specifically, for a denoising trajectory of length $I$,

$$\tau = (\boldsymbol{s}_I, \boldsymbol{a}_I, \boldsymbol{s}_{I-1}, \boldsymbol{a}_{I-1}, \ldots, \boldsymbol{s}_0, \boldsymbol{a}_0), \quad (7)$$

where for each denoising step $i \in \{0, \ldots, I\}$, the state $\boldsymbol{s}_i \in \mathcal{S}$ and action $\boldsymbol{a}_i \in \mathcal{A}$ are determined by the denoiser policy $\boldsymbol{\pi}(\boldsymbol{a} \mid \boldsymbol{s})$. Specifically, the components are defined as follows:

$$\begin{aligned}
\boldsymbol{s}_i &\triangleq (x_{\sigma_i}, \sigma_i, c_v), \\
\boldsymbol{a}_i &\triangleq x_{\sigma_{i-1}}, \\
\boldsymbol{\rho}_0(\boldsymbol{s}_I) &\triangleq (p(c_v), \delta_I, \epsilon), \\
\boldsymbol{P}(\boldsymbol{s}_{i-1} \mid \boldsymbol{s}_i, \boldsymbol{a}_i) &\triangleq (\delta_{x_{\sigma_{i-1}}}, \delta_{\sigma_{i-1}}, \delta_c), \\
\boldsymbol{R}(\boldsymbol{s}_i, \boldsymbol{a}_i) &\triangleq \mathbb{1}_{[i=0]} \cdot r(x_{\sigma_0}), \\
\boldsymbol{\pi}(\boldsymbol{a}_i \mid \boldsymbol{s}_i) &\triangleq p(x_{\sigma_{i-1}} \mid x_{\sigma_i}, c_v).
\end{aligned} \quad (8)$$

Here, $\delta$ is the Dirac delta distribution. The initial state distribution $\rho_0$ is determined by the condition distribution $p(c_v)$, starting with timestep $I$ and Gaussian white noise $\epsilon$. The reward function is only provided at the final denoising step $i = 0$, where $r(x_{\sigma_0})$ is the reward function used to evaluate the quality of the generated output. The transition kernel $\boldsymbol{P}$ describes the deterministic state transitions after an action is taken.

Thus, the problem becomes a policy optimization problem

where we aim to learn a policy that maximizes the reward by optimizing the model's prediction of the visual representation over time. Formally, we define our objective as

$$\mathcal{J}(\boldsymbol{\pi}) = \mathbb{E}_{\tau \sim \boldsymbol{\pi}} \left[ \sum_{i=0}^{I} \boldsymbol{R}(\boldsymbol{s}_i, \boldsymbol{a}_i) \right]. \tag{9}$$

### 3.2. Denoising with SDE-Sampling

In the original VPM pipeline, denoising is performed via a deterministic policy, denoted as Euler Discrete solver , consistent with the deterministic probability-flow ODE framework (Song et al., 2020b). This formulation yields a deterministic denoising path rooted in the initial state. Since this deterministic process provides no stochastic exploration and lacks the tractable transition density $p(x_{\sigma_{i-1}} \mid x_{\sigma_i}, c_v)$ required by policy gradient methods (Black et al., 2023), we instead employ the Euler-Ancestral sampler (Karras et al., 2022). This approach transforms the ODE update into an SDE transition through step-wise Gaussian noise injection. For a transition from $\sigma_i$ to $\sigma_{i-1}$, we sample:

$$x_{\sigma_{i-1}} = x_{\sigma_{i-1}}^{\text{det}} + \sigma_{\text{up}}\epsilon, \quad \epsilon \sim \mathcal{N}(0, \mathbf{I}). \tag{10}$$

Here, the deterministic component is

$$x_{\sigma_{i-1}}^{\text{det}} = x_{\sigma_i} + (\sigma_{\text{down}} - \sigma_i) \frac{x_{\sigma_i} - D_{\theta_v}(x_{\sigma_i}, \sigma_i, c_v)}{\sigma_i}, \tag{11}$$

where $D_{\theta_v}$ is the VPM denoiser network, and the noise level is decomposed into deterministic and stochastic parts:

$$\sigma_{\text{up}} = \sqrt{\sigma_{i-1}^2 \frac{\sigma_i^2 - \sigma_{i-1}^2}{\sigma_i^2}}, \tag{12}$$

$$\sigma_{\text{down}} = \sqrt{\sigma_{i-1}^2 - \sigma_{\text{up}}^2}. \tag{13}$$

This yields an explicit Gaussian transition density:

$$p_\theta(x_{\sigma_{i-1}} \mid x_{\sigma_i}, c_v) = \mathcal{N}\left(x_{\sigma_{i-1}}; x_{\sigma_{i-1}}^{\text{det}}, \sigma_{\text{up}}^2 \mathbf{I}\right). \tag{14}$$

We denote $\boldsymbol{\pi}^{eas}(\boldsymbol{a} \mid \boldsymbol{s})$ as denoiser with the Euler Ancestral sampler

### 3.3. Reinforcement Learning with Hybrid Sampler

In our framework, we employ Group Relative Policy Optimization (GRPO) (Xue et al., 2025) to fine-tune the VPM, enabling it to better capture the complex dynamics of future observations. Our methodology is structured into two primary stages: Rollout with Euler Hybrid sampler and Optimization with verifiable reward.

**Rollout with Euler Hybrid sampler.** For a given conditional input $c_v$, the VPM must produce a group of $G$

---

**Algorithm 1** GRPO Training for Video Prediction Model
______________________________________________________
**Require:** Initial policy $\boldsymbol{\pi}_{\theta_{\text{old}}}$, dataset $\mathcal{D}$, group size $G$
**Ensure:** Optimized policy $\boldsymbol{\pi}_\theta$
1: Initialize $\boldsymbol{\pi}_\theta \leftarrow \boldsymbol{\pi}_{\theta_{\text{old}}}$
2: **for** iteration $t = 1, 2, \ldots, T$ **do**
3:     Sample batch of conditions $c_v = (o, l)$ and expert Future Representations $x_0$ from $\mathcal{D}$
4:     Sample shared initial noise $x_{\sigma_I} \sim \mathcal{N}(0, \sigma_I^2 \mathbf{I})$
5:     **for** $g = 1$ to $G$ **do**
6:         Roll out hybrid sampling: first step SDE, remaining steps ODE $\rightarrow$ obtain $x_{\sigma_0}^g$
7:         Compute reward $r_g$ from $(x_{\sigma_0}^g, x_0)$
8:     **end for**
9:     Compute advantages $A_g$ via group normalization
10:     Update policy parameters $\theta$ by maximizing $J(\theta)$
11:     Update $\boldsymbol{\pi}_{\theta_{\text{old}}} \leftarrow \boldsymbol{\pi}_\theta$ periodically
12: **end for**
______________________________________________________

diverse candidate future representations $\{x^g\}_{g=1}^G$, which is crucial for stable policy optimization. Specifically, we replace the Euler Discrete solver with the Euler Ancestral sampler $\boldsymbol{\pi}^{eas}$, transforming the denoising process from an ODE formulation into an SDE-based sampling procedure. However, introducing stochasticity at all denoising steps leads to severe credit assignment issues: since the entire trajectory shares the same reward signal, the policy may maximize the return by exploiting later denoising actions that are irrelevant to action modeling, rather than genuinely improving the early-step visual representation that is critical for downstream action decision making.

Therefore, we design Euler Hybrid sampler, which applies SDE sampling only at the first denoising step before the action-relevant representation is captured and keeps the remaining steps deterministic:

$$p(\boldsymbol{s}_{0:I-1}, \boldsymbol{a}_{1:I} \mid \boldsymbol{s}_I) = \boldsymbol{\pi}^{eas}(\boldsymbol{a}_I \mid \boldsymbol{s}_I) \prod_{i=1}^{I-1} \delta(\boldsymbol{a}_i - f_{\text{eds}}(\boldsymbol{s}_i)), \tag{15}$$

where $f_{\text{eds}}$ denotes the deterministic Euler Discrete update function that maps the current denoising state to the next latent via a single Euler ODE step. Moreover, this design further improves training stability and efficiency, as limiting stochasticity to a single denoising step preserves the pretrained temporal consistency of the video model and reduces the number of steps involved in backpropagation.

**Optimization with verifiable reward.** Following the stochastic rollouts, which provide a diverse set of candidate samples $g \in \{1, \ldots, G\}$ ending in a denoised latent $x_{\sigma_0}^g$, we evaluate the predicted future representation against the ground truth. Let $x_0$ be the expert future representation obtained by encoding the corresponding expert future clip

with the VAE encoder. For each sampled representation, we compute a verifiable relative latent-consistency reward:

$$r = -\lambda_{L1} \|x_{\sigma_0} - x_0\|_1 + \lambda_{\cos} \frac{x_{\sigma_0} \cdot x_0}{\|x_{\sigma_0}\| \|x_0\|}. \quad (16)$$

We then compute group-normalized advantages:

$$A_g = \frac{r_g - \text{mean}(\{r_1, r_2, \ldots, r_G\})}{\text{std}(\{r_1, r_2, \ldots, r_G\})}. \quad (17)$$

Since the reward is terminal, we apply the same advantage $A_g$ to all stochastic denoising steps within the trajectory.

The GRPO objective is:

$$J(\theta) = \mathbb{E}_{\boldsymbol{a}_{I,g} \sim \boldsymbol{\pi}_{\theta_{\text{old}}}(\cdot|\boldsymbol{s}_{I,g})} \Big[ \frac{1}{G} \sum_{g=1}^{G} \min(\rho_{I,g} A_g,$$
$$\text{clip}(\rho_{I,g}, 1 - \epsilon_c, 1 + \epsilon_c) A_g) \Big] \quad (18)$$

$$\rho_{I,g} = \frac{\boldsymbol{\pi}_\theta(\boldsymbol{a}_{I,g} \mid \boldsymbol{s}_{I,g})}{\boldsymbol{\pi}_{\theta_{\text{old}}}(\boldsymbol{a}_{I,g} \mid \boldsymbol{s}_{I,g})} \quad (19)$$

where $\epsilon_c$ is the clipping threshold that prevents excessively large policy updates (Schulman et al., 2017). By maximizing $J(\theta)$, the VPM is optimized to align its generative distribution with high-reward denoising trajectories that stay consistent with expert dynamics.

# 4. Experiments

In this section, we aim to answer the following questions:

**Q1.** To what extent does our approach improve the quality of visual dynamics modeling?
**Q2.** Do improvements in visual dynamics modeling translate into measurable gains in policy performance?
**Q3.** Through what mechanisms does enhanced dynamics information lead to improved policy performance?
**Q4.** How does **Dyn-VPP** compare with state-of-the-art methods?
Regarding how each component contributes to the overall framework and the real-world experiments, additional results and analyses are provided in the Appendix.

## 4.1. Experimental Setup

**Simulation Setting.** We evaluate on the CALVIN benchmark (Mees et al., 2022) for long-horizon, language-conditioned robotic manipulation. Following the ABC→D protocol, models are trained on environments ABC and evaluated on the unseen environment D, which differs in visual appearance and layout, requiring both long-horizon execution and generalization. We further assess long-horizon capability on L-CALVIN (Fan et al., 2025), which extends

task sequences from 5 to 10 steps for more challenging multi-step evaluation.

**Baseline.** We adopt (Hu et al., 2025a) as our base policy as it is a highly competitive VPM-based VLA with strong visual dynamics modeling capability. Beyond (Hu et al., 2025a), we compare against representative methods from both VLM-based and VPM-based VLAs. For VLM-based VLAs, we include (Black et al., 2025b;a); for VPM-based VLAs, we evaluate both pixel-level and latent-level variants such as (Tian et al., 2025). Moreover, to ensure a fair and comprehensive assessment, we also report results of existing techniques that further strengthen our base policy, highlighting a key advantage of our approach: it improves capability without requiring any architectural modifications.

**Training Details.** We perform post-training on a VPM initialized from the pretrained checkpoint provided by VPP (Hu et al., 2025a), and the training data consists of videos from the Calvin ABC dataset. The VPM is optimized for 1.5k training steps using 64 NVIDIA H20 GPUs. Then, we train the AGM on the Calvin ABC dataset for 10 epochs, which is carried out using 8 NVIDIA H20 GPUs. All evaluation is performed on NVIDIA RTX 5880 GPUs.

## 4.2. Evaluation on Visual Dynamics.

**Metrics.** To assess whether visual dynamics improve with our proposed **Dyn-VPP**, we evaluate both quantitative and qualitative performance. Quantitatively, we measure latent generation accuracy by computing the L1 loss between VPM-generated latents and the ground-truth latents on the validation set. This metric aligns with the training objective and directly reflects optimization quality. Qualitatively, we decode the VPM-generated latents back to pixel space and compare the resulting reconstructions with the corresponding ground-truth images to examine whether the model exhibits (i) action rectification, (ii) planning correction, and (iii) reduced scene hallucination.

**Result Analysis.** As shown in Figure 3, the quality of the learned latent representations, measured in terms of visual dynamics, exhibits an overall improving trend throughout training. Although fluctuations are observed at intermediate stages, the visual dynamics encoded in the latent space become progressively more coherent and structured as training proceeds. In addition, a qualitative comparison between the baseline and predicted future observations demonstrates improved alignment with expert dynamics. Specifically, the optimized VPM produces latent representations that decode into more accurate object poses, spatial relationships, and contact progression, which in turn facilitate more effective action generation.

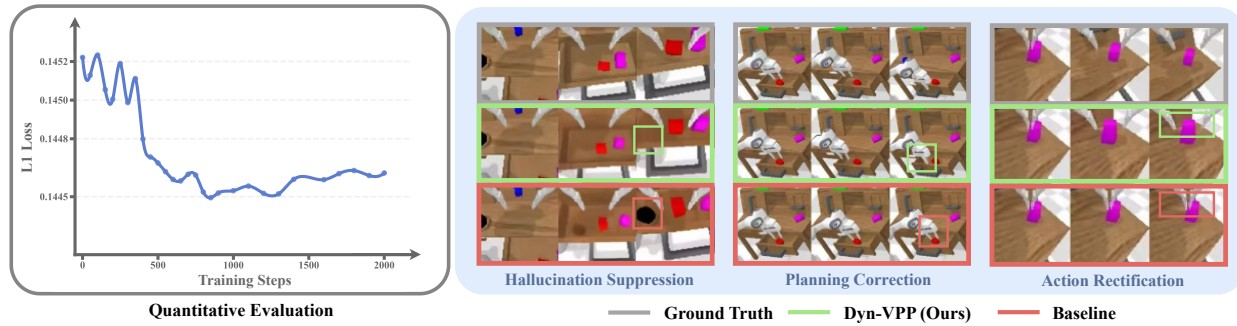

*Figure 3.* Evaluation on Visual Dynamics.

*Table 1.* Effectiveness of improved visual dynamics on CALVIN ABC→D. The table reports task completion in a row (1–5), average trajectory length (Avg. Len), average effective rank (Avg. ER), and average ER ratio (Avg. ERR). Post-training both VPM and AGM achieves the best performance; improvements over the base policy are in **bold**.

| CALVIN ABC→D | Task completed in a row ↑ | | | | | Avg. Len ↑ | Avg. ER ↑ | Avg. ERR ↑ |
|---|---|---|---|---|---|---|---|---|
| | 1 | 2 | 3 | 4 | 5 | | | |
| *Base Policy* | 96.0 | 91.3 | 86.4 | 80.4 | 74.7 | 4.28 | 29.28 | 0.0603 |
| + post-training VPM + original AGM | 96.3 | 91.4 | 87.0 | 82.9 | 77.2 | 4.35 | – | – |
| + post-training VPM + post-training AGM | **98.0** | **94.8** | **91.3** | **88.3** | **83.1** | **4.56** | **43.88** | **0.0814** |

## 4.3. Evaluation on Action Execution

**Metrics.** Here, we primarily evaluate the model's success rate (SR), and in the CALVIN environment we additionally report the average executed trajectory length (Avg.Len).

**Results.** As shown in Table 1, even when the downstream AGM is frozen, optimizing the VPM consistently improves overall performance. Although the gains are modest, this is expected because the downstream policy is not co-optimized. Furthermore, after fine-tuning the VPM with our method, enabling training of the downstream AG leads to substantially larger improvements. These results indicate that unoptimized visual dynamics can misguide downstream policy learning (equivalently, the mapping learned by the inverse dynamics model). Once the visual dynamics are properly optimized, more accurate visual-dynamics representations can be translated into better action dynamics, highlighting the advantage of our approach that directly updates visual dynamics via policy-gradient optimization.

## 4.4. Correlation between Visual Dynamics and Action

**Metrics.** We measure vision–action coupling using the **effective rank** (ER) of the Jacobian $\mathbf{J} = \partial\mathbf{a}/\partial\mathbf{v}$. After SVD with singular values $\{\sigma_i\}$, ER is defined as

$$\mathrm{ER}(\mathbf{J}) = \frac{\left(\sum_i \sigma_i\right)^2}{\sum_i \sigma_i^2}.$$

We report the **Average ER** and the normalized **Average ER Ratio (ERR)** = $\overline{\mathrm{ER}}/\mathrm{ER}_{\max}$, with $\mathrm{ER}_{\max} \leq \min(d_a, d_v)$.

Larger ER indicates richer vision–action coupling.

**Results.** As shown in Table 1, we observe that, compared with the Base Policy, our proposed paradigm achieves substantial improvements in both Avg. ER and Avg. ER Ratio. This indicates that the downstream AGM relies on a larger number of mutually independent visual directions in the learned visual dynamics when generating actions—i.e., the vision–action coupling becomes richer—which in turn leads to fundamental performance gains during the action execution stage.

## 4.5. Evaluation with SOTA Methods

**CALVIN.** Table 2 summarizes comparison with VLM-based and VPM-based VLAs on CALVIN ABC→D. We find that, although our method is built upon VPP, our post-training procedure, which directly optimizes visual dynamics without any architectural modifications, substantially improves VPP's performance and also surpasses approaches based on other model families. Moreover, compared with prior VPP-based extensions such as (Liu et al., 2025b), our method requires neither additional network components nor extra data, yet seamlessly boosts the base model's performance, indicating a fundamental improvement to VPP.

**L-CALVIN.** As shown in Table 3, **Dyn-VPP** yields larger gains on longer-horizon tasks. This further highlights the importance of improving visual dynamics modeling for long-horizon manipulation, as such tasks involve richer and more diverse dynamics over extended time spans.

*Table 2.* Performance on the CALVIN ABC→D benchmark. The table reports task completion in a row (1–5) and average trajectory length (Avg. len) for VLM-based and VPM-based VLAs. **Dyn-VPP** (Ours) achieves the best performance; our method's row is highlighted.

| CALVIN ABC→D | | Task completed in a row ↑ | | | | | Avg. len ↑ |
|---|---|---|---|---|---|---|---|
| | | 1 | 2 | 3 | 4 | 5 | |
| *VLM-based VLA* | OpenVLA (Kim et al., 2024a)$_{(CoRL)}$ | 91.3 | 77.8 | 62.0 | 52.1 | 43.5 | 3.27 |
| | OpenVLA-OFT (Kim et al., 2025)$_{(RSS)}$ | 96.3 | 89.1 | 82.4 | 75.8 | 66.5 | 4.10 |
| | $\pi_0$ (Black et al., 2025b)$_{(RSS)}$ | 93.7 | 83.2 | 74.0 | 62.9 | 51.0 | 3.65 |
| | $\pi_{0.5}$ (Black et al., 2025a)$_{(CoRL)}$ | 92.7 | 84.3 | 76.7 | 68.8 | 61.3 | 3.84 |
| *VPM-based VLA (pixel-level)* | UniPi (Du et al., 2023)$_{(NeurIPS)}$ | 56.0 | 16.0 | 8.0 | 8.0 | 4.0 | 0.92 |
| | SuSIE (Black et al., 2024)$_{(ICLR)}$ | 87.0 | 69.0 | 49.0 | 38.0 | 26.0 | 2.69 |
| | CLOVER (Bu et al., 2024)$_{(NeurIPS)}$ | 96.0 | 83.5 | 70.8 | 57.5 | 45.4 | 3.53 |
| *VPM-based VLA (latent-level)* | GR-1 (Wu et al., 2024)$_{(ICLR)}$ | 85.4 | 71.2 | 59.6 | 49.7 | 40.1 | 3.06 |
| | Vidman (Wen et al., 2024)$_{(NeurIPS)}$ | 91.5 | 76.4 | 68.2 | 59.2 | 46.7 | 3.42 |
| | Seer (Tian et al., 2025)$_{(ICLR)}$ | 94.4 | 87.2 | 79.9 | 72.2 | 64.3 | 3.98 |
| | Seer$_{Large}$ (Tian et al., 2025)$_{(ICLR)}$ | 96.3 | 91.6 | 86.1 | 80.3 | 74.0 | 4.28 |
| | VPP (Hu et al., 2025a)$_{(ICML)}$ | 96.0 | 91.3 | 86.4 | 80.4 | 74.7 | 4.28 |
| | **Tri-VLA** (Liu et al., 2025b)$_{(Arxiv)}$ | **96.8** | **92.4** | **86.8** | **83.2** | **81.8** | **4.37** |
| **Dyn-VPP (Ours)** | | 98.0 | 94.8 | 91.3 | 88.3 | 83.1 | 4.56 |

*Table 3.* Performance on L-CALVIN (long-horizon). The table reports tasks completed in sequence (1–10) and average trajectory length (Avg. Len) under the ABC→D protocol. **Dyn-VPP** (Ours) yields the best results across all task lengths; improvements are highlighted.

| Train→Test | Method | Tasks completed in sequence ↑ | | | | | | | | | | Avg. Len ↑ |
|---|---|---|---|---|---|---|---|---|---|---|---|---|
| | | 1 | 2 | 3 | 4 | 5 | 6 | 7 | 8 | 9 | 10 | |
| ABC→D | OpenVLA | 0.67 | 0.34 | 0.24 | 0.12 | 0.03 | 0.01 | 0.01 | 0.01 | 0.00 | 0.00 | 1.43 |
| | $\pi_0$ | 0.84 | 0.64 | 0.51 | 0.43 | 0.34 | 0.25 | 0.21 | 0.17 | 0.11 | 0.11 | 3.61 |
| | VPP | 0.94 | 0.82 | 0.75 | 0.63 | 0.53 | 0.49 | 0.43 | 0.35 | 0.31 | 0.28 | 5.53 |
| **Dyn-VPP (Ours)** | | **0.97** | **0.89** | **0.81** | **0.75** | **0.71** | **0.64** | **0.61** | **0.51** | **0.45** | **0.39** | **6.73** |

## 5. Related Work

**Video-Action Models.** Video Model-based VLAs represent a more fundamentally novel paradigm than traditional VLM-based VLAs. Recent works on Video Action Models (Hu et al., 2024; Zhang et al., 2025; Pai et al., 2025; Kim et al., 2026; Wang et al., 2025a) have introduced a new direction for VLA modeling by explicitly incorporating future visual prediction into action generation. In contrast, conventional VLM-based VLAs (Gong et al., 2024; Brohan et al., 2022; Zitkovich et al., 2023; Li et al., 2023; Kim et al., 2024b; Song et al., 2025) typically learn a direct mapping from current observations and language instructions to actions, without explicitly modeling the underlying system dynamics. This predictive structure enables the policy to reason over temporal evolution and action consequences, rather than relying solely on static representations.

**Diffusion Models for Robot Control.** Reinforcement learning has recently been integrated with diffusion models to improve generation quality and policy behaviors. DDPO (Black et al., 2023) formulates denoising as a sequential decision process. GRPO-based extensions for generative models—including flow-based variants (Liu et al., 2025a; Xue et al., 2025), hierarchical and structural variants (Ding & Ye, 2025; Li et al., 2025), and generative alignment frameworks (Wang et al., 2025b; Zheng et al., 2025)—further explore combining RL's long-term reward optimization with the expressive power of diffusion models. Our work adapts GRPO principles to optimize video prediction models in latent space, explicitly treating prediction quality as an intermediate objective and strengthening the dynamical prior for downstream VLA policies.

## 6. Conclusion

We introduced **Dyn-VPP**, a video prediction policy optimization method that improves the precision of visual dynamics for manipulation. By treating denoising as a sequential decision process and leveraging a custom Euler Hybrid sampler with a GRPO-based reward, **Dyn-VPP** aligns predicted latents with expert dynamics. Experiments in simulated and real-world settings show consistently improved predictions, better downstream task performance, and enhanced generalization.

# Acknowledgements

This work was supported by the Natural Science Foundation of China under Grant 52308215, Grant 62403401 and the Brain Science and Brain-like Intelligence Technology — National Science and Technology Major Project (Grant No. 2022ZD0208800)

# Impact Statement

**Positive impact.** Our work improves the accuracy of visual dynamics predicted by diffusion-based video models for robot manipulation, without changing model architectures. More reliable future prediction directly supports safer and more capable language-conditioned robots in assistive and domestic settings, logistics and manufacturing, and applications where precise contact and pose matter. Because **Dyn-VPP** is a post-training procedure, it can be applied to existing Vision–Language–Action (VLA) systems without redesign, lowering the barrier for practitioners to obtain better manipulation performance from current video-action models. Improvements in long-horizon and out-of-distribution settings (e.g., L-CALVIN, unseen environments) may also support more robust deployment in non-stationary or diverse environments.

**Potential misuse and limitations.** The same advances in manipulation could be used in autonomous systems where human oversight is limited; developers and deployers should consider domain-specific safeguards and monitoring. Our method relies on expert demonstrations and a hand-designed latent-consistency reward; poor or biased demonstrations or reward design could lead to degraded or inequitable behavior. We do not foresee unique dual-use risks beyond those common to general-purpose robot learning from demonstration.

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

# A. Appendix

## A.1. Evaluation on Core Components.

Next, we provide an ablation analysis of the core components of our approach to validate the effectiveness and rationale of each design module.

*Table 4.* Ablation on post-training steps (CALVIN ABC→D). Success rates and average length for different GRPO step counts. Best performance is achieved at 1400 steps; best values are in **bold**.

| | | Task completed in a row ↑ | | | | | Avg. Len ↑ |
|---|---|---|---|---|---|---|---|
| | | 1 | 2 | 3 | 4 | 5 | |
| *Base Policy* | | 96.0 | 91.3 | 86.4 | 80.4 | 74.7 | 4.28 |
| + | 400 | 97.4 | 93.8 | 90.2 | 87.0 | 81.8 | 4.50 (+0.22) |
| + | 600 | 95.6 | 90.6 | 85.6 | 81.8 | 77.2 | 4.31 (+0.03) |
| + | 1000 | 96.5 | 92.8 | 89.0 | 84.9 | 78.2 | 4.42 (+0.14) |
| **+ 1400 (Ours)** | | **98.0** | **94.8** | **91.3** | **88.3** | **83.1** | **4.56** (+0.28) |

**Post-Training Steps.** From Table 4, we observe that our method reaches a relatively strong performance within only 400 post-training steps. However, the overall training process is somewhat unstable, with occasional performance drops in the middle of training. Despite these fluctuations, our method consistently outperforms the base policy, and achieves its best performance at 1400 steps. This trend reflects the inherent stochasticity of reinforcement learning; importantly, the results at every evaluated checkpoint remain higher than the base policy, demonstrating the robustness and effectiveness of our approach.

*Table 5.* Ablation on optimization algorithm. Comparison of DDPO and GRPO on CALVIN ABC→D. GRPO (Ours) achieves the best performance.

| | | Task completed in a row ↑ | | | | | Avg. Len ↑ |
|---|---|---|---|---|---|---|---|
| | | 1 | 2 | 3 | 4 | 5 | |
| *Base Policy* | | 96.0 | 91.3 | 86.4 | 80.4 | 74.7 | 4.28 |
| + | DDPO | 96.1 | 91.4 | 87.5 | 83.4 | 77.6 | 4.36 (+0.08) |
| **+ GRPO (Ours)** | | **98.0** | **94.8** | **91.3** | **88.3** | **83.1** | **4.56** (+0.28) |

**Optimization Algorithm.** As shown in Table 5, GRPO consistently outperforms DDPO under the same base policy, yielding higher success rates across all horizons and a larger improvement in average trajectory length. We hypothesize that the within-group optimization and clipping scheme in GRPO lead to more stable and informative gradient updates, whereas DDPO is more sensitive to noisy rewards and can underutilize the diversity of candidate rollouts.

**Hybrid Denoising Type.** In Table 6, we verify our motivation for applying SDE to the first step only. This is mainly because applying SDE to later steps can lead to reward

*Table 6.* Ablation on hybrid denoising: SDE at 5 steps vs. 1 step. Applying SDE only at the first step (Ours) yields the best performance and mitigates reward hacking.

| | | Task completed in a row ↑ | | | | | Avg. Len ↑ |
|---|---|---|---|---|---|---|---|
| | | 1 | 2 | 3 | 4 | 5 | |
| *Base Policy* | | 96.0 | 91.3 | 86.4 | 80.4 | 74.7 | 4.28 |
| + | 5-step SDE | 95.9 | 91.9 | 88.1 | 82.3 | 76.4 | 4.34 (+0.06) |
| **+ 1-step SDE (Ours)** | | **98.0** | **94.8** | **91.3** | **88.3** | **83.1** | **4.56** (+0.28) |

hacking: the first-step output is fed directly to the policy, whereas perturbing later steps may encourage the model to exploit the reward in unintended ways.

*Table 7.* Ablation on reward type: pixel-level vs. latent-level reward on CALVIN ABC→D. Latent-space reward (Ours) achieves the best performance.

| | | Task completed in a row ↑ | | | | | Avg. Len ↑ |
|---|---|---|---|---|---|---|---|
| | | 1 | 2 | 3 | 4 | 5 | |
| *Base Policy* | | 96.0 | 91.3 | 86.4 | 80.4 | 74.7 | 4.28 |
| + | Pixel | 96.8 | 91.9 | 88.2 | 83.6 | 78.0 | 4.39 (+0.08) |
| **+ Latent (Ours)** | | **98.0** | **94.8** | **91.3** | **88.3** | **83.1** | **4.56** (+0.28) |

**Reward Type.** For reward computation, we consider two alternatives: computing the reward directly in the latent space, or decoding latents back to the image space and using a pixel-level reward. As shown in Table 7, the latent-space reward achieves better performance. We attribute this to the fact that pixel-level consistency does not necessarily imply more accurate visual dynamics modeling, whereas latent representations are more aligned with the underlying dynamics and task-relevant semantics

*Table 8.* Weight Sensitivity of L1 and Cosine Similarity Rewards (Push Block Right Task)

| Reward Setting | Success Rate (%) ↑ | Improvement |
|---|---|---|
| Base | 56.9 | — |
| L1 only | 79.2 | +22.3 |
| Cosine only | 80.6 | +23.7 |
| L1:Cos = 1:0.2 | 88.9 | +32.0 |
| L1:Cos = 0.2:1 | 86.1 | +29.2 |
| L1:Cos = 1:1 | 88.9 | +32.0 |

**Reward weighting.** We additionally conducted a sensitivity study on the push block right task to evaluate the effect of different weightings between the L1 and cosine similarity rewards. Overall, the results indicate that both reward terms are individually effective, and that their combination

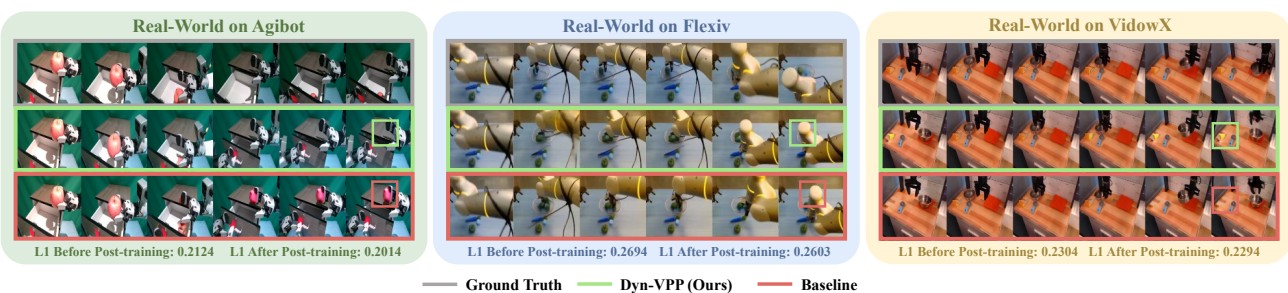

*Figure 4.* Visualization results.

is generally more beneficial than either term alone. The performance variation across different weight ratios is small, suggesting that Dyn-VPP is reasonably robust to the choice of this hyperparameter. In particular, balanced weightings, as well as settings that slightly favor L1, tend to provide the strongest performance. This observation supports the use of the 1:1 weighting in the main paper as a stable default configuration.

*Table 9.* Alternative Latent-Space Reward Forms on the *Push Block Right* Task

| Reward Type | Success Rate (%) ↑ | Improvement |
|---|---|---|
| Base | 56.9 | — |
| L1 | 79.2 | +22.3 |
| L2 | 80.6 | +23.7 |
| Cosine Similarity | 80.6 | +23.7 |
| Latent Flow | 81.7 | +24.8 |

**Reward choices.** We also evaluated several alternative single-term latent-space rewards on the push block right task. The results show that all considered reward forms consistently improve over the base model, indicating that Dyn-VPP is not overly sensitive to the exact reward formulation. This suggests that, for Dyn-VPP, the central factor is the optimization of the video predictor through a latent-space objective, rather than the specific algebraic form used to define the reward. Overall, these findings provide further evidence of the method's robustness to reward design choices.

### A.2. Evaluation on Real-World.

**Real-World Setting.** Real-world experiments are conducted on the Agibot Genie 01 dual-arm robot. The platform provides 14 DoF across both arms and 2-DoF grippers on each end effector, enabling dexterous bimanual coordination and stable grasping for tabletop daily manipulation (e.g., single-arm grasping, bimanual cooperation, and multi-object handling). Perception is provided by an Intel RealSense D455 depth camera mounted on the head, offering a third-person global view of the tabletop and arm motions,

and two Intel RealSense D405 depth cameras mounted on the wrists for close-range, first-person observations of grasp geometry and contact. We consider three real-world tasks: (1) grasping a specified target (apple) in clutter, (2) single-arm precise pick-and-place of a plate onto a central saucer, and (3) bimanual simultaneous grasp-and-place of two bottles onto a central plate. For each task, we collect 200 high-quality demonstrations with synchronized multi-view visual observations and robot state/action logs.

**Analysis.** As shown in Figure 5, our **Dyn-VPP** consistently outperforms the base policy in real-world scenarios, demonstrating the effectiveness and robustness of the proposed approach.

### A.3. Training Details

We perform post-training on a Video Prediction Model (VPM) initialized from the pretrained checkpoint provided by VPP. The training data consists of videos from the Calvin ABC-D dataset, which contains diverse long-horizon manipulation trajectories. In this stage, the VPM is optimized using reinforcement learning for 1.5k training steps with 64 NVIDIA H20 GPUs. Training is conducted in a distributed data-parallel manner.

After VPM post-training, we train the AGM on the Calvin ABC-D dataset for around 10 epochs, using 8 NVIDIA H20 GPUs. Unless otherwise specified, all training is implemented in PyTorch under a distributed setting and executed on NVIDIA RTX 5880 GPUs.

Unless otherwise stated, we use a group size of $G = 8$ for group-wise policy optimization. The PPO-style clipping parameter is set to $\epsilon_{\text{clip}} = 0.2$. For optimization, we adopt a learning rate of $1 \times 10^{-4}$ during fine-tuning, and a smaller learning rate of $1 \times 10^{-6}$ for GRPO-based optimization. The reward function combines an $L_1$ distance term and a cosine similarity term, with both reward weights set to $1.0$. The batch size is fixed to $8$ across all experiments.

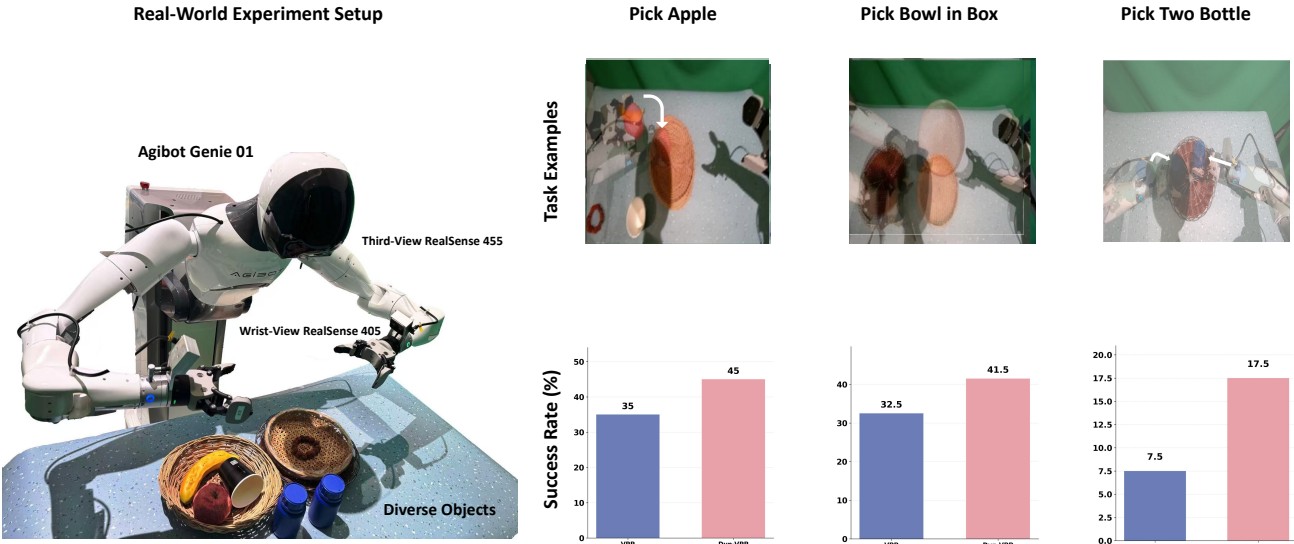

*Figure 5.* Real-World Experiments.

## A.4. Visualization

We provide qualitative visualizations to demonstrate the effectiveness of our post-training approach for Action Prediction Models (APM) across multiple environments and robotic platforms. In addition to experiments in the Calvin environment and the real-world Agibot, we also perform experiments on Bridge and the real-world Flexiv dual-arm robot.

As shown in Figure 4, our method consistently improves action prediction and manipulation quality. We observe improvements in three key aspects. First, the model generates fewer implausible or irrelevant motions compared to the pretrained VPM, effectively suppressing hallucinations. Second, high-level action sequences better align with task goals, leading to more coherent planning. Third, individual motor commands are refined, resulting in smoother and more precise execution. These visualizations highlight that post-training enhances both the semantic understanding and kinematic fidelity of the model, demonstrating robust generalization across different robots and environments.

## B. The Use of Large Language Models (LLMs)

To enhance the readability and coherence of this paper, we employed large language models to assist in refining the writing.

## C. Reproducibility Statement

We will release training and evaluation code, configuration files, and scripts to reproduce all results in the paper. The release will include (i) dataset preprocessing for expert demonstrations, (ii) the exact noise schedule and sampler settings (Euler Discrete, Euler–Ancestral, and Euler Hybrid), (iii) GRPO training code with group normalization and clipping, and (iv) evaluation scripts and metric computation. We will also provide the random seeds used for each experiment, hardware and software versions, and instructions to download any required pretrained checkpoints (e.g., VAE and text encoder) with corresponding licenses.

