# OpenReview forum: "Dyn-VPP: Video Prediction Policy Optimization for Improved Visual Dynamics"
_ICML.cc/2026/Conference — ICML 2026 regular_

### Official Review · Reviewer_DN7Q · 2026-03-10

**Soundness:** 3
**Presentation:** 3
**Significance:** 2
**Originality:** 2
**Overall Recommendation:** 4
**Confidence:** 3

**Summary:**

During the training of video-action models, existing methods are usually trained via likelihood objectives that emphasize global plausibility. This paper proposes Dyn-VPP, a video prediction policy optimized for the accuracy of precision-critical factors using GRPO. The authors show Dyn-VPP surpass baselines in CALVIN and real world experiments.

**Compliance With Llm Reviewing Policy:**

Affirmed.

**Final Justification:**

The rebuttal addressed most of my concerns. I think this paper provides a good enhancement on VPP via GRPO post training. My original concerns about reward function designs and motivation claims are also addressed. So I raise my score to 4.

**Key Questions For Authors:**

Questions:

1. In Figure 3a, what does the x-axis represent? In Figure 3b, what do the three rows correspond to?

2. The reward defined in Equation 16 is based on the assumption that latent consistency provides a better control-related signal than pixel information. How can one quantitatively prove that better latent consistency actually yields precision-critical visual dynamics for manipulation?

3. The performance gain might also originate from the use of GRPO for diffusion optimization, rather than the claim that it "guarantees precision-critical visual dynamics."


4. Typos at Line 216 “figure” -> “Figure”. L280 Baseline section cite -> citet.

**Strengths And Weaknesses:**

Strengths:

1. The paper attempts to solve a critical problem in video-action models: existing models are optimized using pixel-level loss (or likelihood-based loss), which does not necessarily align with downstream policy performance.

2. The proposed method is post-training and can directly optimize the existing approaches.


Weaknesses:

1. The reward defined in Equation 16 is based on the assumption that latent consistency provides a better control-related signal than pixel information, which is not immediately obvious.

2. The latent vector used in the diffusion usually contains all information since we need to reconstruct the full pixel space based on that. But the reward contains only a L1 loss and cosine similarity applied to the entire vector. Further explanations are needed.

Overall I appreciate the technical contribution and I believe using GRPO as a post-training approach can further improve VPP performance. I think this is a good paper in general. But I think what i'm not convinced is the mismatch between the motivation and the method part.

---

> ### Author Rebuttal · Authors · 2026-03-29
>
> We sincerely appreciate your time and effort in reviewing our paper! Thank you for your support in our work!
> ___
>
> **W1: Why latent consistency provides a better control-related signal than pixel information?**
>
> Thank you for pointing this out. We agree that this distinction should be stated more clearly in the main paper. We have added an ablation in Appendix Table 7 (also showed in our responses to Reviewer mDQ6 Q2) comparing our latent-space reward with a pixel-space reward (MAE + LPIPS on decoded images). The results show that the **latent-space reward is clearly more effective than the pixel-space reward**.
>
> We believe the reason is that **pixel similarity is more easily affected by appearance factors** such as texture, lighting, and background, which are not necessarily strongly correlated with action generation. This is also consistent with the intuition in JEPA-VLA [1]: rather than forcing the model to focus on pixel changes, it is more beneficial to encourage representations that **capture the underlying task state**.
> ___
>
> **W2: Why the reward is designed with L1 and Cosine similarity?**
>
> As discussed above in W1, the ablation study shows the **advantage of optimization in latent space**, while the effective-rank analysis in Table 1 suggests that latent visual features **still contain substantial action-irrelevant information**. Our design is therefore intended to emphasize the control-relevant **structured information** in the latent representation: the **L1 term** constrains absolute deviation, and the **cosine term** encourages overall structural consistency.
>
> More importantly, we would like to emphasize that we also examined the sensitivity of Dyn-VPP to different reward choices (see our responses to Reviewer mDQ6 Q3), and the results show that the effectiveness of the method does **not** depend on one highly specialized reward design. In this sense, **how to optimize the video predictor is more important than the exact algebraic form of the reward**.
> ___
> **Q1: Ambiguity in Figure 3.**
>
> Thank you for the careful reading. In Figure 3a, the x-axis denotes the **training steps during Dyn-VPP post-training**. In Figure 3b, the three rows correspond to the visualizations of **GT**, **Dyn-VPP**, and **Baseline**, respectively. We will clarify this more explicitly in the figure caption in the revised version to avoid ambiguity.
> ___
>
> **Q2: How can one quantitatively prove that better latent consistency actually yields precision-critical visual dynamics for manipulation?**
>
> We agree that this claim should be made more carefully. First, in our framework, the AGM essentially plays the role of an **inverse dynamics model (IDM)**: it takes the predictive precision-critical visual features as input to generate actions **where even subtle deviations can significantly affect action accuracy**. Therefore, the most direct quantitative test of whether latent consistency truly benefits manipulation is to examine downstream control performance. Empirically, we observe that the improved latent consistency translates consistently into **higher success rate / Avg. Len**.
>
> Second, we also have added an experiment (see our responses to Reviewer mDQ6 Q1), which provides a more direct quantitative analysis of improving visual dynamics. The results show that better latent consistency is associated with more accurate **task-relevant visual dynamics** (**spatial alignment** , **perceptual quality**, and **motion consistency**), rather than merely superficial visual improvements.
>
> Frankly, the evaluation of visual dynamics is still limited. We will therefore explore more comprehensive evaluation protocols in the future.
>
> ___
> **Q3: The performance gain might also originate from the use of GRPO for diffusion optimization, rather than the claim that it guarantees precision-critical visual dynamics.**
>
> In our method, **GRPO itself does not automatically guarantee more accurate visual dynamics**. This is supported by the comparison between 1-step SDE and 5-step SDE (see our response to Reviewer BiPD W1). Both are optimized under the same GRPO-based diffusion optimization framework, yet their final performance is clearly different. Interestingly, 5-step SDE achieves **slightly better latent consistency**, but **significantly worse downstream performance**.
>
> Therefore, our view is that the gain does **not** mainly come from GRPO improving the overall diffusion generation. Instead, the key reason is that **our design makes the visual dynamics for AGM more precise**, by focusing optimization on the task-relevant predictive representation.
>
> ___
>
> **Q4: Typos and format issues.**
>
> Thank you for pointing out the typos and citation-format issues. We will correct them in the revised version and carefully proofread the manuscript for related issues.
> ___
> We hope our clarification can solve all your concerns!
> ___
> [1] JEPA-VLA: Video Predictive Embedding is Needed for VLA Models

---

> > ### Author Rebuttal · Reviewer_DN7Q · 2026-04-04
> >
> > I'm satisfied with the author's response. I would like to raise my score to 4.

---

> > > ### Author Response · Authors · 2026-04-04
> > >
> > > Dear Reviewer DN7Q,
> > >
> > > Thank you very much for your acknowledgment and kind feedback. We are very glad that our rebuttal has satisfactorily addressed your concerns, and we sincerely appreciate your positive assessment of our work.
> > >
> > > We noticed that the official review score may not yet have been updated. If appropriate and convenient, we would be very grateful if your revised assessment could also be reflected there.
> > >
> > > Thank you again for your time, consideration, and supportive feedback.
> > >
> > > Best regards,
> > > Authors

---

### Official Review · Reviewer_mDQ6 · 2026-03-11

**Soundness:** 3
**Presentation:** 3
**Significance:** 3
**Originality:** 2
**Overall Recommendation:** 4
**Confidence:** 3

**Summary:**

This paper proposes Dyn-VPP, a post-training framework for diffusion-based video action models that improves downstream manipulation by directly optimizing the video prediction module (VPM) for more accurate visual dynamics. The key idea is to reinterpret diffusion denoising as an MDP, optimize it with a GRPO-style objective, and use a hybrid sampler that injects stochasticity only at the first denoising step. The paper argues that standard likelihood-based VPM training optimizes global plausibility rather than the precision-critical dynamics needed for control, and shows that improving these latent dynamics leads to better downstream action generation and long-horizon task execution in simulation and real-world manipulation.

**Compliance With Llm Reviewing Policy:**

Affirmed.

**Key Questions For Authors:**

- The paper argues that Dyn-VPP improves precision-critical visual dynamics such as object pose, fine-grained spatial relations, and contact timing. However, the main quantitative evaluation of visual dynamics appears to rely primarily on latent L1 loss. Are there any more direct metrics or analyses that specifically measure these dynamics-related factors?
- The appendix compares GRPO against DDPO, but it is unclear how much of the gain comes from GRPO-based reward optimization itself, as opposed to simply continuing to fine-tune the VPM on the same dataset. Could the authors clarify whether a plain supervised fine-tuning baseline was considered, and if so, how it compares?
- It would be helpful if the authors could clarify whether alternative reward formulations were explored beyond the latent-space L1 + cosine similarity objective, and how sensitive Dyn-VPP is to this choice.

**Limitations:**

- The method depends on access to expert future clips and a reward defined from latent agreement with those futures.
- The paper focuses on a specific video-action-model pipeline, VPP, so it is not yet clear how broadly the approach transfers across different architectures and domains.

**Strengths And Weaknesses:**

Strengths:

- Clear motivation: the paper identifies an important mismatch between likelihood-based video prediction training and control-relevant future dynamics.
- The proposed framing of denoising as policy optimization is conceptually reasonable, and adapting GRPO-style post-training to latent video prediction showed high gains in both visual dynamics and policy.


Weaknesses:

- While the downstream gains are convincing, the paper could more cleanly isolate improvements in the VPM itself from benefits obtained after retraining the AGM. In particular, it remains unclear how much of the improvement is specific to GRPO-based reward optimization, as opposed to continued optimization on the same dataset.
- The evaluation of visual dynamics would be stronger with more direct metrics for manipulation-relevant precision (e.g., object pose, fine-grained spatial relations, or contact timing), rather than relying primarily on latent L1 loss and qualitative decoding.

---

> ### Author Rebuttal · Authors · 2026-03-29
>
> We sincerely appreciate your time and effort in reviewing our paper! Thank you for your support in our work!
> ___
> **Q1: Are there any more direct metrics or analyses that specifically measure these dynamics-related factors?**
>
> Thank you for the suggestion. We agree that latent-space L1 alone is insufficient to fully support the claim of improved precision-critical visual dynamics. Moreover, key factors such as object pose and fine-grained spatial relations are **inherently difficult to measure directly from unlabeled real-world videos**. We therefore additionally conduct a more direct **quantitative analysis of generated videos on the real robot platform**.
>
>
> | Metric | VPP | Dyn-VPP (Ours) |
> |---|---:|---:|
> | Mask IoU ↑ | 0.200 | 0.244|
> | MSE ↓ | 0.035 | 0.033 |
> | LPIPS ↓ | 0.322 | 0.308 |
> | RAFT optical flow error ↓ | 22.826 | 21.597 |
>
> Here, Mask IoU is computed by first segmenting the robot arm in the generated videos using Grounded SAM 2, and then measuring the average overlap with the ground-truth masks.  Compared with latent L1 alone, these metrics provide more direct evidence from three complementary perspectives: **spatial alignment** (Mask IoU), **perceptual quality** (MSE / LPIPS), and **motion consistency** (RAFT flow error). All of them consistently support that our method improves **task-relevant visual dynamics**.
> ___
>
> **Q2&W2: Could the authors clarify the isolated benefits from retraining AGM and SFT?**
>
> We showed in Table 1 that the gains **come from improved VPM visual dynamics with AGM frozen**, while AGM retraining further amplifies the improvement with improved visual representation.
>
> We additionally included a **plain supervised fine-tuning (SFT)** baseline. We ran this additional experiment on the first 20% of the test set (200 cases). The results are as follows:
>
> | Method | 1 | 2 | 3 | 4 | 5 | Avg. Len ↑  |
> |---|---:|---:|---:|---:|---:|---:|
> | Baseline | 94.5 | 88.1 | 82.1 | 79.1 | 71.6 | 4.2 |
> | Baseline + SFT (12k steps) | 92.4 | 85.8 | 81.2 | 78.2 | 73.1 | 4.1 |
> | Dyn-VPP | 98.5 | 96.1 | 91.7 | 89.2 | 82.4 | 4.6 |
>
> These results show that **continued supervised fine-tuning on the same data cannot explain the gains of Dyn-VPP**: SFT does not provide consistent improvements over the baseline, whereas Dyn-VPP substantially outperforms both the baseline and SFT across all metrics.
> ___
>
> **Q3: It would be helpful if the authors could clarify whether alternative reward formulations were explored.**
>
> We have indeed explored other reward forms, and our conclusion is that while the specific reward formulation does affect the final performance, **the effectiveness of Dyn-VPP does not rely on one highly specialized reward design**.
>
> First, in Appendix Table 7, we compare a **pixel-space reward** (MAE + LPIPS) with our latent-space reward, and find that the pixel-space reward is clearly less effective. This suggests that, for video action models, directly optimizing in pixel space does not effectively improve the task-relevant visual dynamics, whereas latent-space supervision is better aligned with this problem setting.
>
> Second, we also tested several alternative single-term rewards in latent space on the single task push block right (trained for 200 steps on a single GPU, evaluated on 72 test cases):
>
> | Reward | Success Rate (%) | Improvement |
> |---|---:|---:|
> | Base | 56.9 | — |
> | L1 | 79.2 | +22.3 |
> | L2 | 80.6 | +23.7 |
> | Cosine | 80.6 | +23.7 |
> | Flow (latent) | 81.7 | +24.8 |
>
> These results show that different latent-space rewards all lead to stable improvements, indicating that Dyn-VPP is not overly sensitive to the exact reward form. In other words, **how to optimize the video predictor is more central than the exact algebraic form of the reward**.
> ___
> **L1: The method depends on access to expert future clips and a reward defined from latent agreement with those futures.**
>
> Thank you for pointing this out. We agree that the current method relies on access to expert future clips. We would like to clarify that this setting does **not** introduce supervision **beyond the standard training paradigm of video-action models**, whose training data already consists of expert future clips without any further annotation.
>
> More broadly,  the reward can also be defined using other signals that correlate with task-relevant visual dynamics, such as learned value functions, success predictors, or environment-derived metrics when available. Exploring alternative-supervision settings is an important direction for future work.
> ___
> **L2: It is not yet clear how broadly the approach transfers across different architectures and domains.**
>
> We suppose that our method is **in principle transferable** to such settings. Due to the rebuttal length limit, we refer the reviewer to our responses to **Reviewer GVDz (W1 and W2)**, where we provide a more detailed analysis of this point.
> ___
> We hope our clarification can solve all your concerns!

---

> > ### Author Rebuttal · Reviewer_mDQ6 · 2026-04-02
> >
> > Thank you for the rebuttal. The authors have addressed my concerns, and I will keep my positive score.

---

> > > ### Author Response · Authors · 2026-04-02
> > >
> > > Dear Reviewer mDQ6,
> > >
> > > Thank you very much for your acknowledgment and for your positive assessment of our work. We sincerely appreciate your time and thoughtful consideration throughout the review process. We are very glad that our rebuttal has adequately addressed your concerns.
> > >
> > > If you feel it appropriate after considering our clarifications, we would be truly grateful if you could kindly reconsider the score. In any case, we deeply appreciate your helpful feedback and encouragement.
> > >
> > > Best regards,
> > >
> > > Authors

---

### Official Review · Reviewer_GVDz · 2026-03-13

**Soundness:** 3
**Presentation:** 3
**Significance:** 3
**Originality:** 3
**Overall Recommendation:** 5
**Confidence:** 4

**Summary:**

This paper study a empirical question in video action model. Since it's time consuming to de-noise a  whole video, many video action model will using the representations in the first forward step. The author propose a method to directly optimize the representation in initial denoising steps which is more correlated to the final manipulation success rate. Overall, I think the paper explore a meaningful direction and execution is good.

**Compliance With Llm Reviewing Policy:**

Affirmed.

**Final Justification:**

I appreciate the clarification from the authors. My main concerns have largely been resolved, and I do not have further questions. I have therefore raised my score to 5.

**Key Questions For Authors:**

1. see weakness

**Limitations:**

No. Author should add some limitation parts in the last section.

**Strengths And Weaknesses:**

Strength:
1. The motivation is clear. The paper articulates the mismatch between likelihood-based diffusion training and action-centric precision requirements quite well.
2. The proposed method is also reasonable. Forming denoising as an MDP and using RL-style post-training is a natural extension of DDPO-like ideas to video prediction for control.
3. The improvement in experiments is also convincing. With only training target change, it improve the original VPP[1] baseline.

Weakness:
1. The method appears to be tailored to a specific class of video-action models. In particular, it seems to rely on models that use only a small number of denoising steps. More importantly, the algorithm seems closely tied to the VPP-style architecture, where video prediction and action generation are separated. It is unclear whether the approach would extend to other types of video-action models in which video generation and action prediction are jointly modeled within a single network [2,3].

2.  It is also unclear how necessary this method would be if one directly denoises the final video through multiple steps, as in Cosmos-Policy [4]. In that setting, would the proposed method still provide a meaningful advantage?

[1] Video prediction policy: A generalist robot policy with predictive visual representations
[2]  Large Video Planner Enables Generalizable Robot Control
[3] World action models are zero-shot policies
[4] Cosmos Policy: Fine-Tuning Video Models for Visuomotor Control and Planning

---

> ### Author Rebuttal · Authors · 2026-03-29
>
> We sincerely appreciate your time and effort in reviewing our paper! Thank you for your support in our work!
> ___
> **W1: It is unclear whether the approach would extend to other types of video-action models in which video generation and action prediction are jointly modeled within a single network?**
>
> Thank you for this question. We agree that the current implementation in our paper is built on a latent-level video-action model where the video prediction model (VPM) and the action generation model (AGM) are relatively decoupled. Therefore, for architectures in which video generation and action prediction are fully unified within a single network, the applicability of our method indeed requires further clarification.
>
> Our view is that the core idea of Dyn-VPP does **not** rely on the existence of explicit module boundaries, but rather on **whether there exists an optimizable process for future visual prediction / denoising**. If a joint model still contains, either explicitly or implicitly, a diffusion-based denoising process responsible for generating future video or future visual representations, then our method should in principle remain applicable: this denoising process can still be formulated as a sequential decision process, and reward-based post-training can be applied to the part responsible for future visual prediction.
>
> That said, for such unified architectures, **additional care would be needed** in handling the coupling introduced by shared parameters during training. For example, one may need to avoid unstable interference with the action prediction pathway, or selectively optimize only the generation branch or intermediate layers most related to predictive visual dynamics.
>
> Overall, we believe the idea is **in principle transferable** to jointly modeled video-action architectures, but it would require additional architecture-specific adaptation and training design. We view this as an important direction for future work.
> ___
> **W2: It is also unclear how necessary this method would be if one directly denoises the final video through multiple steps, as in Cosmos-Policy [4]. In that setting, would the proposed method still provide a meaningful advantage?**
>
> We believe that our method remains meaningful even in settings where the model performs multi-step denoising directly to generate the final video. The key issue is **not** whether the model outputs the final video itself, but whether its training objective directly constrains the **manipulation-relevant visual dynamics**.
>
> More specifically, even if a model, such as Cosmos-Policy, generates the final video through multi-step denoising, its training is **still typically dominated by reconstruction- or likelihood-based objectives**. Such objectives can encourage the generated video to be globally realistic and temporally coherent, but they do not necessarily prioritize the factors that are most critical for manipulation. From this perspective, our method is not in conflict with models that generate the final video; rather, it can be viewed as a **complementary post-training mechanism** for existing multi-step video denoising models.
>
> Of course, if a model already introduces very strong supervision signals explicitly aligned with manipulation success during joint training, then the marginal gain from our method may become smaller. However, we believe the method would still remain beneficial, and we plan to validate this more systematically in future work.
> ___
> **L1: Author should add some limitation parts in the last section.**
>
> Thank you for the suggestion. We agree that the paper would benefit from a clearer discussion of its limitations. The current study is evaluated on a limited set of architectures, and its generalizability remains to be explored. The assessment of visual dynamics is also preliminary and requires a more comprehensive framework. We will add this to the final section in the revised version.
> ___
> Thank you again for your questions! We will keep polishing the work to make it meet the highest standard!

---

> > ### Author Rebuttal · Reviewer_GVDz · 2026-04-02
> >
> > I appreciate the clarification from the authors. My main concerns have largely been resolved, and I do not have further questions. I have therefore raised my score to 5.

---

> > > ### Author Response · Authors · 2026-04-02
> > >
> > > Dear Reviewer GVDz,
> > >
> > > Thank you for your acknowledgment and kind feedback. We sincerely appreciate your time and consideration. We are glad that our response helped clarify your concerns, and we will carefully take your suggestions into account in future revisions of the paper.
> > >
> > > Best regards,
> > >
> > > Authors

---

### Official Review · Reviewer_BiPD · 2026-03-18

**Soundness:** 3
**Presentation:** 3
**Significance:** 2
**Originality:** 2
**Overall Recommendation:** 3
**Confidence:** 3

**Summary:**

The paper introduces Dyn-VPP, a post-training framework designed to improve the accuracy of visual dynamics in diffusion-based video action models. The authors identify an "objective mismatch" in current Vision-Language-Action (VLA) systems: while standard video predictors are trained for global likelihood (plausibility), they often lack the precision in object poses and contact timing necessary for successful robotic manipulation. Dyn-VPP treats the multi-step diffusion denoising process as an MDP optimized via Group Relative Policy Optimization (GRPO)

**Compliance With Llm Reviewing Policy:**

Affirmed.

**Final Justification:**

Rebuttal addressed many concerns. I still believe the work lacks novelty to meet the bar of ICML.

**Key Questions For Authors:**

How sensitive is the performance to the weighting of the $L_{1}$ vs. Cosine similarity rewards?

**Limitations:**

Yes.

**Strengths And Weaknesses:**

- The methodology is technically grounded, effectively adapting GRPO—originally for LLMs—to the diffusion denoising process
- The paper is well-structured and clearly motivated.
- The work builds heavily on DDPO and GRPO. While the application to latent video dynamics is novel, the underlying RL mechanisms are established in other generative domains. For this reason, the work is marginally novel.

---

> ### Author Rebuttal · Authors · 2026-03-29
>
> We sincerely appreciate your time and efforts in reviewing our paper! Thank you for your support on our work!
> ___
> **W1: While the application to latent video dynamics is novel, the underlying RL mechanisms are established in other generative domains. For this reason, the work is marginally novel.**
>
> We understand the reviewer’s concern. We would like to clarify that, although our method is inspired by influential works such as DDPO and GRPO, the RL algorithm itself is not our main contribution; in fact, our framework is largely **algorithm-agnostic**. Our novelty lies in the following **two** contributions.
>
> 1.**We identify a new core problem which has never been systematically solved.**
>
> Different from prior work that mainly improves the action model, we point out an **objective mismatch** between the training objective of the video prediction model and the precision-critical visual dynamics required for manipulation. Based on this, we establish a closed loop showing **how VPM post-training leads to downstream policy improvement**.
>
> 2.**Applying RL to robotic video action models requires new domain-specific designs, rather than directly reusing existing generative-model post-training methods.**
>
> We provide clarification from the following two aspects.
>
> **Reward design.** As shown in Table below (i.e., Appendix Table 7), simply applying an RL algorithm with pixel reward (MAE & LPIPS) that has been validated in other generative domains does **not** yield the best performance in robotics. Instead of using human-preference or pixel-level rewards commonly adopted in image/video generation, we explore a latent-consistency reward that better aligns with expert trajectories in latent space and provides more control-relevant visual dynamics. This result suggests that **RL methods effective in other domains cannot directly transfer to robotic manipulation without a reward tailored to control-relevant representations**.
>
> | Method | 1 | 2 | 3 | 4 | 5 | Avg. Len ↑ |
> |---|---:|---:|---:|---:|---:|---:|
> | Base Policy | 96.0 | 91.3 | 86.4 | 80.4 | 74.7 | 4.28 |
> | + Pixel | 96.8 | 91.9 | 88.2 | 83.6 | 78.0 | 4.39 (+0.08) |
> | + Latent (Ours) | 98.0 | 94.8 | 91.3 | 88.3 | 83.1 | 4.56 (+0.28) |
>
>
> **Credit assignment.** As shown in Table below (i.e., Appendix Table 6), even with the same RL optimization framework, introducing stochasticity at multiple denoising steps—as in standard DDPO-style designs—does **not** work as well for robotic video action models. The model may exploit later denoising steps to improve the terminal reward without improving the early visual representation actually used by the downstream policy. Our Euler Hybrid sampler is specifically designed to address this robotics-specific credit-assignment issue, and the ablation verifies that **without this design, RL algorithms proven effective in other generative settings do not translate into equally strong gains in robot manipulation**.
>
> | Method | 1 | 2 | 3 | 4 | 5 | Avg. Len ↑ |
> |---|---:|---:|---:|---:|---:|---:|
> | Base Policy | 96.0 | 91.3 | 86.4 | 80.4 | 74.7 | 4.28 |
> | + 5-step SDE | 95.9 | 91.9 | 88.1 | 82.3 | 76.4 | 4.34 (+0.06) |
> | + 1-step SDE (Ours) | 98.0 | 94.8 | 91.3 | 88.3 | 83.1 | 4.56 (+0.28) |
> ___
> **Q1: How sensitive is the performance to the weighting of the L1 vs. Cosine similarity rewards?**
>
> Thank you for this question. We additionally conducted a sensitivity study on a single task, push block right, to evaluate the effect of different weights for the L1 and Cosine rewards. The experiment was run for 200 training steps on a single A100 GPU, and evaluated on 72 test cases. The results show that **Dyn-VPP is not highly sensitive to the relative weighting of the L1 and Cosine terms**, indicating good overall robustness.
>
> | Reward Setting | Success Rate (%) | Improvement  |
> |---|---:|---:|
> | Base | 56.9 | — |
> | L1 only | 79.2 | +22.3 |
> | Cosine only | 80.6 | +23.7 |
> | L1 : Cos = 1 : 0.2 | 88.9 | +32.0 |
> | L1 : Cos = 0.2 : 1 | 86.1 | +29.2 |
> | L1 : Cos = 1 : 1 | 88.9 | +32.0 |
>
> These results suggest two main points:
>
> 1.**Each reward term is effective on its own.** Even when using only a single reward component, the performance is already substantially better than the base model.
>
> 2.**Using both rewards together is generally better than using either one alone**, and the performance remains similar across different weight ratios. In particular, relatively balanced settings or those slightly favoring L1 perform the best.
>
> Overall, these results indicate that the **1:1 weighting used in our paper is a stable default choice that does not require careful tuning**, and that **Dyn-VPP is reasonably robust to this hyperparameter**.
> ___
> Thank you again for your time and effort in reviewing our work! We hope our clarification can solve all your concerns, and we are always ready to answer any further questions!

---

> > ### Author Rebuttal · Reviewer_BiPD · 2026-04-03
> >
> > I acknowledge your response. I appreciate the engineering effort of the work, however I still find the novelty limited. I will keep the score as is.

---

> > > ### Author Response · Authors · 2026-04-04
> > >
> > > Thank you for the reviewer’s comments. We would like to further clarify that our work is **not** a simple transfer of existing RL mechanisms to another generative task. We believe the novelty of this paper lies in **three aspects**:
> > >
> > > **1.** this is the **first work to systematically study post-training for Video Action Models (VAMs)**;
> > > **2.** rather than starting from method transfer, we start from **failure analysis** and identify an **objective mismatch** in VAMs for manipulation that has long existed but has not been systematically articulated;
> > > **3.** we further show that post-training designs effective in other generative domains do **not** directly apply to VAMs, which motivates our **robotics-specific designs**.
> > >
> > > We will elaborate on details as follows.
> > >
> > > **1. First systematic study of VAM post-training**
> > >
> > > To the best of our knowledge, this is the first work to study post-training for **Video Action Models (VAMs)**. Existing VAM-related works mainly focus on model architecture design, or pretraining/fine-tuning of the action model, but they do not truly address a fundamental question: **what kind of visual dynamics representation is actually more valuable for downstream robotic manipulation in the VAM setting?** We believe this question must be answered first in order to make more effective use of limited robot data and to drive further progress in this area.
> > >
> > > **2. Identifying the objective mismatch in VAMs through failure analysis**
> > >
> > > Our work begins with an analysis of failure cases in existing VAMs. We observe that many task failures in current VAMs can be traced back to errors in video generation: the predicted results often remain globally plausible visually, yet small deviations are significantly amplified in downstream action decisions, eventually causing failures in grasping, contact, or trajectory execution. Based on this observation, we further realized that the issue is not simply that the model “does not generate well enough,” but that the training objective of existing diffusion-based video predictors is not aligned with the control-relevant visual dynamics actually required for manipulation. In other words, **model training mainly optimizes global plausibility and distribution matching, whereas downstream robot policies depend on precision-critical visual dynamics**. This is precisely the core problem we identify: an **objective mismatch** between the training objective of video prediction models and the control-relevant visual dynamics required for manipulation.
> > >
> > > **3. Tailored post-training designs for control-relevant visual dynamics in VAMs**
> > >
> > > After identifying this issue, we **did not simply reuse post-training methods from other generative domains**. Instead, we investigate what kinds of post-training designs are truly effective for VAMs in robotic manipulation. We first experimented with existing designs such as pixel-level rewards. However, our experiments show that such methods do not work well in the VAM setting. We believe one key reason is that, for robotic manipulation, **low-level pixel consistency is not the core information that action generation truly relies on**. Similarity in texture, color details, or local appearance may reflect visual closeness of generated results, but is often of limited value for action decisions. In contrast, robot policies rely more on **higher-level, structured semantic and dynamic information**. Based on this understanding, we further study the relationship between video prediction and action generation, and analyze their coupling and information redundancy. From this insight, we propose the **latent consistency reward**, which more accurately constrains the visual dynamics that truly matter for manipulation. In addition, we find that **representations at early denoising steps in VAMs carry robot dynamics information more effectively**. If stochasticity is injected across multiple steps as in standard DDPO-style methods, the model may improve terminal reward by exploiting later steps, without actually improving the early representations that are most critical for control. Based on this insight, we design the **Euler Hybrid sampler**, which introduces stochasticity only at the first step, so that optimization focuses more directly on the visual dynamics most relevant to downstream manipulation.
> > >
> > > Overall, we believe the contribution of this work is **not** a simple reuse of existing RL mechanisms, nor can it be summarized as mere engineering effort. The above three key contributions are reflected not only in the method design, but also in our rethinking of the core problem in VAMs, our analysis of failure mechanisms, and our systematic understanding of the specific needs of robotic manipulation. **We believe these contributions go beyond engineering implementation and provide a new perspective on how video action models can better serve robot manipulation.**
> > >
> > > We sincerely appreciate your time and consideration.
> > >
> > > Best regards,
> > >
> > > Authors

---

### Decision · Program_Chairs · 2026-04-30

**Decision:**

Accept (regular)

**Comment:**

This paper addresses an interesting and important problem: current diffusion models for video prediction only yield visually plausible futures without considering much about the precision-critical visual dynamics required for robotic manipulation. Multiple reviewers praised the paper's clear motivation, the soundness of the methodology, and the practicality of the proposed post-training framework.

During the rebuttal phase, the authors successfully addressed various reviewer concerns, providing reaonable answers regarding the evaluation of visual dynamics and the reward formulation (latent vs. pixels). Ultimately, three out of the four reviewers expressed that their concerns were satisfied and voted favorably for the submission.

The single dissenting reviewer's remaining concern centered on limited algorithmic novelty, characterizing the work as primarily an engineering effort. However, the authors clarified that their core contribution is not the RL algorithm itself, but rather identifying and addressing the "objective mismatch" in Vision-Language-Action (VLA) models. Furthermore, their domain-specific design choices produce concrete improvements over the baseline, demonstrating that simply transferring existing RL methods is ineffective for robotics.

The AC finds the authors' response persuasive. The merits and substantial technical contributions of this paper outweigh the novelty concerns. Therefore, the AC recommends accepting this submission.